# COVID-19, Renin-Angiotensin System and Endothelial Dysfunction

**DOI:** 10.3390/cells9071652

**Published:** 2020-07-09

**Authors:** Razie Amraei, Nader Rahimi

**Affiliations:** Department of Pathology, School of Medicine, Boston University Medical Campus, Boston, MA 02118, USA

**Keywords:** SARS-CoV-2, endothelial dysfunction, ACE2, endothelial cell injury, CD209L, L-SIGN

## Abstract

The newly emergent novel coronavirus disease 2019 (COVID-19) outbreak, which is caused by SARS-CoV-2 virus, has posed a serious threat to global public health and caused worldwide social and economic breakdown. Angiotensin-converting enzyme 2 (ACE2) is expressed in human vascular endothelium, respiratory epithelium, and other cell types, and is thought to be a primary mechanism of SARS-CoV-2 entry and infection. In physiological condition, ACE2 via its carboxypeptidase activity generates angiotensin fragments (Ang 1–9 and Ang 1–7), and plays an essential role in the renin-angiotensin system (RAS), which is a critical regulator of cardiovascular homeostasis. SARS-CoV-2 via its surface spike glycoprotein interacts with ACE2 and invades the host cells. Once inside the host cells, SARS-CoV-2 induces acute respiratory distress syndrome (ARDS), stimulates immune response (i.e., cytokine storm) and vascular damage. SARS-CoV-2 induced endothelial cell injury could exacerbate endothelial dysfunction, which is a hallmark of aging, hypertension, and obesity, leading to further complications. The pathophysiology of endothelial dysfunction and injury offers insights into COVID-19 associated mortality. Here we reviewed the molecular basis of SARS-CoV-2 infection, the roles of ACE2, RAS signaling, and a possible link between the pre-existing endothelial dysfunction and SARS-CoV-2 induced endothelial injury in COVID-19 associated mortality. We also surveyed the roles of cell adhesion molecules (CAMs), including CD209L/L-SIGN and CD209/DC-SIGN in SARS-CoV-2 infection and other related viruses. Understanding the molecular mechanisms of infection, the vascular damage caused by SARS-CoV-2 and pathways involved in the regulation of endothelial dysfunction could lead to new therapeutic strategies against COVID-19.

## 1. Introduction

The severe acute respiratory syndrome (SARS) epidemic, which was caused by SARS-CoV, emerged in 2002–2003 in southern China and soon spread to Europe and North America [1,2,3]. A novel coronavirus, SARS-CoV-2, was originally found in patients with severe pneumonia in Wuhan, China at the end of 2019 [4,5]. The disease caused by SARS-CoV-2 was named as COVID-19 [6,7]. SARS-CoV-2 was able to spread rapidly and efficiently, which may account for its significant lethality compared to related viruses such as SARS-CoV and MERS-CoV. Since December 2019, COVID-19 has spread around the world, causing a pandemic that threatens global public health with high mortality in humans and resulted in near complete halt in economic and social activities around world. As of today (8 July 2020), SARS-CoV-2 has infected more than 11 million people and killed over 544,000 worldwide (data compiled by Johns Hopkins University). The major leading cause of mortality in patients with COVID-19 is respiratory failure from acute respiratory distress syndrome (ARDS) [1]. Other causes of mortality include multiorgan failure involving heart and the kidneys [8,9]. However, individuals with comorbidities such as hypertension, diabetes, and obesity have worst outcomes and, in general, men are more affected than women [10]. 

Endothelial dysfunction is an important component of a number of human diseases that also represents the common denominator of all COVID-19 co-existing conditions such as hypertension, diabetes, and obesity which are major contributing factors for COVID-19-related deaths. Consistent with this hypothesis, other clinical manifestations of COVID-19 include cardiac injury [9] and hypercoagulability as measured by an increased in D dimer and Von Willebrand factor (VWF) levels [11,12,13,14]. A recent study found that nearly 72% of non-survivors of COVID-19 had evidence of hypercoagulability [15]. In addition, inflammatory markers including, C-reactive protein, ferritin, interleukin (IL)-6, IP-10, MCP1, MIP1A, and TNF-α all were elevated in COVID-19 patients [16]. Numerous factors such as inflammation could contribute to the hypercoagulability in COVID-19 patients. However, pulmonary and peripheral endothelial cell injury due to direct SARS-CoV-2 infection is a likely scenario, as endothelial cell injury can strongly activate the coagulation system [17] and aggressive immune response could further augment endothelial dysfunction. Considering that Von Willebrand factor (VWF) levels is significantly elevated in COVID-19 patients (529 U/dL compared to 100 U/dL, normal) further supports the hypothesis of SARS-CoV-2 induced endothelial dysfunction or damage [13]. VWF is a circulating adhesive glycoprotein that is secreted by endothelial cells and platelets and its levels is elevated in vasculitis, inflammation, aging [18], and diabetes [19], conditions that are all associated with endothelial dysfunction. VWF activates platelets leading to platelet aggregation [20], acts as a carrier of coagulation factor VIII, and contributes to blood coagulation [21]. Moreover, VWF is a key player in vasculature system including, regulation of angiogenesis and vascular permeability. The chest X-ray or computed tomography (CT) scan found extensive vascular damage as well as evidence of respiratory distress in COVID-19 patients leading to conclusion that COVID-19 could be a disease that primarily damages the vascular endothelium [22]. The interaction between comorbidity factors, SARS-CoV-2, and vascular dysfunction/injury is shown (Figure 1). 

## 2. Novel Severe Acute Respiratory Syndrome Coronavirus-19

The name of coronavirus is derived from the Latin word “corona” meaning crown. It is named as such because of the large spike protein (S protein) molecules on the virions surface that creates a crown-like shape. In general, coronaviruses are classified into at least four major genera, α, β, δ, and γ [23,24]. SARS-CoV and SARS-CoV-2 belong to the β-genus [25,26] and are considered zoonotic pathogens [27] that can infect various species, particularly mammals and birds. Coronaviruses contain an envelope, a helical capsid, and a single-stranded, positive-sense RNA genome with a length of 27–32 kb [25,28]. The whole genome of SARS-CoV-2 was recently sequenced [29]. The 5′ end of the viral genome encodes two polyproteins (e.g., pp1a and pp1ab), which are cleaved by two viral proteases, 3C-like protease (3CLpro) and papain-like protease (PLpro). This leads to generation of 16 non-structural proteins, such as RNA-dependent RNA polymerase (RdRp), which together form the replication complex. The 3′ end of the genome of SARS-CoV-2 encodes four essential structural proteins including, spike (S), envelope (E), matrix/membrane (M), and nucleocapsid (N), along with a set of accessory proteins [29,30]. Remarkably, the SARS-CoV-2 spike protein is highly similar to that of SARS-CoV, as the amino acid sequence identity between SARS-CoV and SARS-CoV S-proteins is about 76% [29]. Spike proteins are essential for cellular entry and infection of coronaviruses. Angiotensin-converting enzyme2 (ACE2) was identified as a functional receptor for hCoV-NL63 [31], SARS-CoV [32], and SARS-CoV-2 [33]. However, the spike protein encoded by MERS-CoV, despite its high similarly to the spike protein of SARS-CoV and SARS-CoV-2 viruses, recognizes CD26 (also known as dipeptidyl peptidase 4, DPP4) as a receptor for cellular entry and infection [34]. 

## 3. ACE2 Peptidase Activity and Signal Transduction

ACE2 is a type-I transmembrane receptor with a catalytic extracellular domain, a single transmembrane domain, and a cytoplasmic carboxyl domain (Figure 2A). ACE2 gene is mapped to the X-chromosome and encodes for 805 amino acid long ACE2 protein [35]. While the extracellular domain of ACE2 consists of zinc metallopeptidase catalytic site and spike binding domain (Figure 2B) [36], the carboxyl terminal domain of ACE2 displays a significant homology to collectrin protein [37], which regulates amino acid re-absorption in the kidney [38,39]. Although the functional role of the C-terminal collectrin homology domain of ACE2 remains largely unknown, it was reported that calmodulin interacts with the C-terminal of ACE2 and inhibits its ectodomain shedding [40,41]. Calmodulin is a ubiquitously expressed protein in mammalian cells and plays major roles in many calcium-mediated cellular processes, such as the regulation of the activity of a large number of enzymes (e.g., protein kinases and phosphatases), ion channels, and aquaporins [42]. 

The key physiological function of ACE2 is associated with its metalloprotease activity which plays a critical role in the regulation and metabolism of RAS circulating peptides by serving as a counter regulatory mechanism to oppose the effects of angiotensin II (Ang II) generated by ACE. ACE2 catalyzes reactions by utilizing zinc, which is coordinated by highly conserved histidine residues within the active site to facilitate nucleophilic attack on the carbonyl bond of the substrate. Additionally, the two histidine (H) residues located within the HEXXH motif and a glutamate residue (E) are involved in coordinating the zinc ion [43]. In addition to zinc, ACE2 activity also is regulated by chloride ions [44]. A recent structural analysis of ACE2 bound with inhibitor, MLN 4760 [45], demonstrated a large ‘hinge-bending’ motion, in which the catalytic subdomains I and II of the peptidase domain exhibit open-to-close transitions. In general, ACE2 can catalyze polypeptides with a substrate preference for hydrolysis between proline, and a hydrophobic or basic carboxyl terminal residue [44]. While ACE is known to convert Ang I (1–10) to the potent vasoconstrictor Ang II (Ang 1–8), ACE2 cleaves Ang I (1–10) to generate Ang 1–9 peptide (Figure 3A). Furthermore, ACE2 metabolizes Ang II (Ang 1–8) to generate Ang 1–7 with a significantly higher efficiency than converting Ang 1–10 to Ang II (Ang 1–9) (Figure 3A). The peptides generated by ACE2 peptidase activity (i.e., Ang1–9 and Ang 1–7) bind to and activate the G-protein-coupled receptor (GPCR), Mas (also called MAS1 and proto-oncogene Mas), with a major vasoprotective function [46,47]. 

GPCR Mas activation leads to stimulation of major signaling pathways including the activation of phospholipase A (PLA) to generate arachidonic acid (AA), phosphoinositide 3 kinase (PI3K)/AKT axis, which activates eNOS by phosphorylation at serine 1177 and activation of phospholipase C leading to stimulation of intracellular calcium [48,49]. Activation of these pathways together regulate vasodilation, and anti-fibrosis and anti-inflammatory responses in endothelial cells (Figure 3B). In contrast, the peptide, Ang 1–8 produced by the action of ACE binds to and activates GPCRs, Angiotensin-2 type 1 receptor 1 (AT1, also called AGTR1), and Angiotensin II type 1 receptor 2 (AT2 also called AGTR2). Activation of AT1R results in the activation of a plethora of kinases (e.g., JAK, p38, MAPK, p38) that modulate vasoconstriction, fibrotic remodeling, and inflammation (Figure 3B). However, activation of AT2R stimulates various phosphatases (e.g., PTP and PP2A) [50,51] leading to vasodilation and growth inhibition (Figure 3B). 

Another important aspect of the regulation of ACE2 is its unusual phosphorylation on the extracellular domain (Figure 2A). Phosphorylation of ACE2 at Ser680 appears to be important for ACE2′s stability. Phosphorylation of Ser680 inhibits ubiquitination of ACE2 and hence increases its surface expression. AMP-activated protein kinase (AMPK) was identified as a kinase responsible for phosphorylation of ACE2 at Ser680 [52]. In general, secretory proteins and the ectodomain of cell surface receptors are not expected to be subject to phosphorylation as protein kinases are not present at the extracellular environment. However, evidence suggests that ACE2 is phosphorylated in the cytoplasm before it reaches to cell surface. Furthermore, it is thought that under certain conditions such as treatment with metformin or AICAR, ACE2 is predominantly localized in the cytoplasm [52], which may explain its phosphorylation at the extracellular domain. In agreement with the functional role of Ser680 phosphorylation on ACE2 activity, the knock-in phosphomimetic-S680 ACE2 (S680D) mice were resistant to pulmonary hypertension [52], suggesting the AMPK-mediated regulation of the vasoprotective function of ACE2. Therefore, other ACE2 functions such as its interaction with SAR-CoV-2-sipke could potentially be targeted through modulation of AMPK via small molecule kinase inhibitors. 

## 4. Angiotensin-Converting Enzyme 2 is a Viral Recognition Receptor

As stated above, aside from its conserved peptidase activity, ACE2 also acts as a moonlighting protein by serving as a functional receptor for SARS-CoV and SARS-CoV-2 [32,33,53,54]. The affinity of spike protein of SARS-CoV-2 for ACE2 appears to be significantly higher compared to SARS-CoV [54], suggesting that SARS-CoV-2 has more efficient cellular entry and infection rate compared to SARS-CoV. This functional role of ACE2 in viral infection appears to be independent of its peptidase activity as catalytic inactive mutants of ACE2 still act as functional receptor for the SARS-CoV [32,53]. It has been suggested that interaction of viral spike protein with an intact and cell surface localized ACE2 leads to the internalization of ACE2 via clathrin-dependent and -independent endocytosis pathways [55,56], which presumably facilitate SARS-CoV and SARSCoV-2 entry into host cells resulting in infection. Furthermore, it appears that the soluble spike protein also can induce the internalization of ACE2. Another important aspect of coronaviruses entry into host cells is their ability to induce ectodomain shedding153 of ACE2 which is required for viral cellular entry and replication [57,58]. Consistent with the SARS-CoV-mediated internalization and ectodomain shedding of ACE2, it has been demonstrated that the SARS-CoV spike protein can trigger downregulation of ACE2 expression in lung tissue and in cell culture [53]. 

Coronavirus-mediated internalization and ectodomain shedding of ACE2 could be in part mediated by the protease activity of disintegrin and metalloproteinase domain-containing protein 17 (ADAM17) and transmembrane protease serine 2 (TMPRSS2). Indeed, a study suggested that TMPRSS2 enables human SARS-CoV infection via two apparently independent mechanisms. TMPRSS2 can cleave ACE2, which might facilitate viral uptake [59]. TMPRSS2 can also cleave spike glycoprotein of SARS-CoV and SARS-CoV-2 which activates the spike protein for cathepsin L-independent host cell entry [60,61]. Additionally, it appears that TMPRSS2 competes with ADAM17 for ACE2 processing. However, only cleavage by TMPRSS2 appears to promote spike protein driven cellular entry [59]. The underlying differences in the outcome of TMPRSS2-mediated versus ADAM17 processing of ACE2 are yet to be elucidated. 

It has been proposed that recognition of ACE2 by coronaviruses is mediated via the trimeric spike protein (i.e., activated spike protein) by binding to a hydrophobic pocket of ACE2 catalytic domain. This results in the internalization and ectodomain shedding of ACE2 and initiates the fusion of virus particles and host cells [58,62,63]. Remarkably, the structural analysis of SARS-CoV and SARS-CoV-2 spike protein-bound ACE2 revealed that the catalytic active site of ACE2 is not inhibited by the spike protein [62,64]. 

## 5. Viral–Host Interaction Activates Spike Protein through Proteolytic Activation

Enveloped viruses such as SARS-CoV-2 access host cells by membrane fusion through either the plasma membrane or an endosomal membrane [65]. Activation of SARS-CoV-2 spike protein through proteolytic cleavage is required for host cell entry [54]. Therefore, an understanding of the envelope protein activation by proteolytic cleavage is central to gaining insights to viral pathogenesis and development of potential therapeutics. The receptor binding domain (RBD) of SARS-CoV-2 spike protein presents the most critical feature of this protein which is located at the C-terminus of the S1 subunit and binds to ACE2 with a high affinity [54]. Mechanistically, it is proposed that the binding of spike protein with ACE2 induces some form of conformational changes, predisposing the spike protein to activation for membrane fusion. Once activated, a fusion peptide at the N-terminus of the S2 subunit inserts into a target host cell membrane via a mechanism that is poorly understood. Finally, the N- and C-terminal heptad repeats within the S2 subunit undergo conformational changes, forming the six-helix-bundle structure [66]. 

Whittak and colleagues proposed a two-step sequential protease cleavage model for the activation of SARS-CoV and MERS-CoV spike proteins [67]. Depending on the virus strains and cell types, spike proteins can be cleaved by one or several host proteases, such as furin, cathepsins, transmembrane protease serine protease-2 (TMPRSS-2), TMPRSS-4, and human airway trypsin-like protease (HAT), which generates receptor binding (S1) and fusion (S2) domains [60,67,68,69,70]. The S1 subunit contains a receptor binding domain (RBD), which is responsible for recognition and binding to the cell surface receptor. The S2 subunit is the “stalk” of the spike protein contains other key features necessary for membrane fusion. It could be argued that multiplicity in the proteases involved in the activation of spike proteins enables these viruses to readily adapt to the new host environment. However, the mechanistic explanation for the role of these proteases in the viral entry to host cells is not fully understood. Another unique aspect of spike protein is that the proteolytic cleavage that leads to membrane fusion can occur both at the interface of the receptor binding, S1 and S2 domains (S1/S2), as well as in a position adjacent to a fusion peptide within S2 subunit [71,72]. It also appears that coronaviruses can use alternative systems for viral entry. For example, MERS-CoV spike protein can be cleaved during protein synthesis, and subsequently the virus can enter host cells via the endosomal pathway in a cathepsin-dependent mechanism. Alternatively, MERS-CoV can enter target cells using the peptidase activity of TMPRSS2 or TMPRSS4 [68,73]. 

## 6. Role of Endothelial Dysfunction in SARS-CoV-2 Infections: Circulating RAS

While endothelial cells play critical roles in the regulation of vascular homeostasis, endothelial dysfunction is associated with pathogenesis and progression of cardiovascular, renal, metabolic, and infectious disease [74,75]. Endothelial dysfunction was initially described as a precursor of atherosclerosis by the imbalance between bioavailability of vasodilators and endothelium-derived vasoconstrictor substances [76]. However, several additional mechanisms and features have been identified including reduced NO production and increased pro-inflammatory, pro-coagulatory and proliferative responses that lead to development of atherogenesis [77,78,79]. Furthermore, some aspects of endothelial dysfunction are interrelated. For instance, oxidative stress is associated with inflammatory responses in diabetes [80] and it can induce inflammation in conditions such as nonalcoholic fatty liver disease [81]. 

There is a strong evidence for complex association between viral infections, inflammatory processes, and endothelial cells. Some viruses such as Human herpes virus 8 (HHV-8) and Hantavirus predominantly infect endothelial cells and increases vascular permeability [82,83,84]. During Hantavirus infection, inhibition of αvβ3 integrins and increase of the sensitivity of endothelial cells to vascular endothelial growth factor (VEGF) have been shown to increase vascular leakage [85]. However, other viruses such as dengue virus targets dendritic cells, monocyte and macrophages promoting cytokine secretion, which in turn activates endothelial cells and induces vascular leakage [84,86]. Dengue virus nonstructural protein 1 (NS1) disrupts endothelial barrier function and inoculation of mice with NS1 alone causes both plasma leakage and production of pro-inflammatory mediators [87,88]. Alteration of endothelial cell function have been extensively studied in the pathogenesis of HIV. Endothelial cells in liver sinusoids, human umbilical veins, and brain microvessels are permissive for HIV infection [89,90,91,92]. Derangement of endothelial function and integrity is mediated either directly by interaction of HIV proteins with endothelial cells or via cytokine secretion through inflammatory cascades [93,94]. The correlation between viro-immunological state of AIDS and the plasma biomarkers of endothelial injury such as Intracellular Adhesion Molecule 1 (ICAM-1), Vascular Cell Adhesion Molecule 1 (VCAM-1), and E-selectin are well-established [95,96]. For instance, HIV-positive patients have higher concentrations of soluble VCAM-1 that is also a predictive biomarker of the disease progression and prognosis [97]. 

The Renin-Angiotensin system (RAS) is a complex cascade of vasoactive peptides controlling the maintenance of blood pressure, tissue perfusion, and extracellular volume via its two pressor and depressor pathways [98,99] (Figure 3A,B). SARS-CoV-2 entry via ACE2 causes downregulation of membrane-bound ACE2 and concurrent loss of catalytic activity of ACE2 in the RAS system [100]. Therefore, SARS-CoV-2 by reducing Ang (1–7) levels could shift the balance towards the pressor arm of the RAS, which could lead to deterioration of cardiovascular homeostasis in COVID-19 patients. Accordingly, ACE2-deficient mice exhibit a variety of complications in cardiovascular system including, increased blood pressure [101], endothelial dysfunction [102], and cardiac structural defects [35]. 

Interestingly, ACE2 expression reduces with aging [103], which predicts a lower rate of SARS-CoV-2 infection in older adults. However, initial studies from China and Italy indicated that the majority of COVID-19 patients were elderly [104,105]. In contrast, in more recent reports from South Korea, most SARS-CoV-2 infections occurred in individuals aged 20–29 years [106]. The discrepancies of age distribution in COVID-19 cases may be explained by differences in testing strategies and other factors. Data from countries with expansive testing programs can reflect more about the association between ACE2 expression levels and higher SARS-CoV-2 infection rate in younger adults. However, reports from the populations with limited testing resources or clinically diagnosed cases do not provide adequate and accurate information about the rate of SARS-CoV-2 infection in different age groups. 

## 7. Role of Endothelial Dysfunction in SARS-CoV-2 Infections: Local RAS

Hypertension, cardiovascular disease, diabetes, and obesity are the most prevalent co-morbidities in COVID-19 patients [107,108,109,110]. The crude fatality rate of COVID-19 patients without any documented comorbidities was 0.9% compared to 10.5% for patients with cardiovascular disease and 7.3% for patients with diabetes [110]. Furthermore, these comorbidity conditions track closely with age, which appears to be the strongest predictor of COVID-19-related death. People with age 45 and higher are more likely to die from COVID-19 compared to a younger age group. Aging is accompanied by complex structural and functional modifications of the vasculature, which ultimately leads to endothelial and smooth muscle cells dysfunction. The ability of aged endothelial cells to produce NO and respond to agonist and mechanical stimuli are significantly reduced [54]. Considering that endothelial dysfunction plays a pivotal role in the pathogenesis of these diseases, exploring the role of endothelial dysfunction could provide further insight into pathogenesis of COVID-19. Furthermore, the emerging data indicate that SARS-CoV-2 also can induce vascular damage [111,112], suggesting that pre-existing endothelial dysfunction combined with the direct assault of SARS-CoV-2 on vascular system may account for a high mortality of COVID-19 patients. 

One possible molecular explanation of these clinical observations could be dysregulation of local RAS system due to SARS-CoV-2 infection. The components of the RAS system have been detected in specific organs such as the heart, lung, and liver, which function through autocrine and paracrine mechanisms independent of circulating RAS [113,114]. The organ-based RAS system plays a specific role in injury/repair response, inflammation, and fibrogenesis pathways [115,116]. For instance, in an acute lung injury model induced by acid aspiration, loss of ACE2 in mice resulted in significantly increased vascular permeability in lungs which is a hallmark of acute lung injury/ARDS in humans [117]. Therefore, SARS-CoV-2 binding to and downregulation of ACE2 is expected to cause the loss of ACE2 protective function in local RAS system of lung which is independent of the ongoing viral infection. The heart is another major tissue-specific RAS organ. Administration of the ACE2 activator (e.g., Diminazene aceturate) is reported to attenuate ischemia-induced cardiac injury, increase circulating endothelial progenitor cells, and restore normal balance of cardiac RAS system in rats. Interestingly, viral RNA and reduced expression of ACE2 in the heart was detected in SARS patients autopsies [118] which may also explain the reported cardiac injuries in COVID-19 cases. A recent study found that ACE2 is expressed in human pericytes and heart failure increases ACE2 expression [119], suggesting that SARS-CoV-2 could predispose COVID-19 patients to cardiac injuries due to loss of cardio-protective function of ACE2 or patients with heart failure have a higher risk of SARS-CoV-2 infection and subsequent cardiac damages. Taken together, the current data suggests that SARS-CoV-2 can pose several challenges to systemic circulation as well as vasculature of lung and heart via modulating ACE2 activity. Although, mechanistic studies in this context are needed to identify high risk individuals and develop potential therapies, exploring alternative routes in the vascular system and other target organs is necessary.

## 8. Cell Adhesion Molecules as Potential Receptors for SARS-CoV2 Host Recognition

To initiate infection, viruses must cross the host-cell plasma membrane, which presents a formidable barrier to cell entry. To overcome this barrier, viruses have evolved multiple different entry mechanisms. This is often accomplished by glycoproteins on the surfaces of host cells that mediate virus attachment and entry. Following attachment, the glycoprotein responsible for mediating host cell entry is activated and becomes fusogenic (e.g., SARS-COV2 spike protein activation and interaction with ACE2). To date, ACE2 is known as the main glycoprotein utilized by SARS-CoV2 for cellular entry. However, recently, several other receptors including, CD209L (L-SIGN), CD209 (DC-SIGN) [120], Neuropilin receptors (NRPs) [121,122], and CD147/Basigin [123] were reported to facilitate SARS-CoV-2 entry. Additionally, a survey of virus–receptor interaction indicates that a requirement for more than one receptor molecule or class is not uncommon and many viruses employ multiple mechanisms for attachment [124,125,126]. Cell adhesion molecules (CAMs) are among the most common receptors exploited by diverse viruses for cell entry [125] (Figure 4). 

The immunoglobulin-like CAMs (IgSF CAMs) superfamily is one of the most common groups of CAMs employed by viruses. IgSF CAMs such as coxsackievirus-adenovirus receptor (CAR) [127], junctional adhesion molecule A (JAM-A) [129,130], intercellular adhesion molecule 1 (ICAM 1/CD54) [128], and poliovirus receptor (PVR/CD155) [131] are known to interact with multiple different viruses, but whether they also interact with coronaviruses remains unknown. In addition to dipeptidyl peptidase 4 (DPP4 also called CD26) [141], MERS-CoV spike protein is known to interact with sialic acid (SA) with particular preference for α2,3-linked SA over α2,6-linked SA [142]. Additionally, MERS-CoV also employs carcinoembryonic antigen-related cell adhesion molecule 5 (CEACAM5), a member of IgSF CAMs, for viral cell entry [132]. Similarly, murine coronavirus interacts with CEACAM6 [143] and porcine hemagglutinating encephalomyelitis coronavirus (PHE-CoV) interacts with IgSF CAM, neural cell adhesion molecule (NCAM) via spike protein [144].

Two particular CAMs that are extensively exploited by various viruses for cell entry are the C-type lectin domain family 4 member M (CLEC4M, also known as L-SIGN and CD209L) and dendritic cell-specific ICAM-3 grabbing non-integrin (DC-SIGN, also called CD209). CLEC4M serves as an attachment receptor for Ebolavirus [133], Hepatitis C virus [134], human coronavirus 229E [135], Human cytomegalovirus/HHV-5 [136], influenza virus [137], West-Nile virus [136], HIV [145], and Japanese encephalitis virus [138] (Figure 4). It has been shown that SARS-CoV can bind to CD209 and CD209L [136,139,140] and CD209L knockout mice were significantly less susceptible to SARS infection [146]. A recent study demonstrated that CD209L is highly expressed in the lung and kidney epithelial and endothelial cells and mediates SARS-CoV-2 entry and infection [120].

## 9. Therapeutic Targets for Treatment of COVID-19

In the face of worldwide pandemic, the development of a vaccine and re-purposing of the previously FDA-approved antiviral drugs have been prioritized for rapid deployment. However, there are multiple other mechanisms, which also represent Achilles heels of SARS-CoV-2, could be exploited. For example, given that SARS CoV-2 cellular entry requires spike protein proteolytic activation, which involves peptidases such as TMPRSS2, furin, and cathepsins, drugs can be developed against these host proteases. Indeed, a recent study demonstrated that TMPRSS2 inhibitor, Camostat mesylate [147] reduces SARS-CoV-2-spike-driven entry into lung epithelial cells [54]. Similarly, since the ectodomain shedding of ACE2 is required for viral entry and peptidase activity of enzymes such as ADAM17 and TMPRSS2 are associated with the ectodomain shedding of ACE2, blocking the ectodomain shedding of ACE2 offers a unique therapeutic opportunity against COVID-19. Another important aspect of development of therapeutic targets against SARS-CoV-2 is inhibition of SARS-CoV-2′s own peptidases such as 3C-like protease (3CLpro) and papain-like protease (PLpro), which are responsible for generation of almost 16 non-structural viral proteins. A recent report has identified α-ketoamide inhibitor as a potential agent to block 3CLpro activity and hence to inhibit SARS-CoV-2 RNA synthesis [148].

Furthermore, targeting the cytoplasmic domain of ACE2 via modulation of calmodulin activity, which regulates the ACE2 ectodomain seeding could be included in the toolbox for the development of anti-viral drugs. For example, multiple calmodulin inhibitors such as the napthalenesulfonamide derivatives W-7/W-13, trifluoperazine, and calmidazolium could be investigated against SARS-CoV-2. More importantly, biological targets such as monoclonal antibodies or soluble proteins corresponding to the extracellular domains of ACE2 or spike protein remain promising therapeutic strategies against SARS-CoV-2. Considering that most of the mortality associated with COVID-19 occurs among people with underlying vascular endothelial dysfunction, in principal, pharmacological interventions that aims to increase nitric oxide (NO) bioavailability could alleviate some of the symptoms associated with COVID-19. Additionally, the effects of drugs such as angiotensin-converting enzyme inhibitors, angiotensin AT1 receptors blockers, angiotensin-(1–7), sphingosine-1-phosphate, β-blockers, calcium channel blockers, endothelial NO synthase enhancers, phosphodiesterase 5 inhibitors, and even cholesterol-reducing statin drugs [149] against SARS-CoV-2 can be explored. It should be noted that concerns were initially raised whether drugs such as ACE inhibitors and angiotensin-receptor blockers [150], could increase the risk of SARS-CoV-2 infection by increasing the expression of ACE2 [151,152]. However, the previous studies on human and animal studies failed to demonstrate an increase in the expression of ACE2 in response to ACE inhibitors and angiotensin-receptor blockers [153,154]. Moreover, a recent study supports the beneficial effects of ACE inhibitors in COVID-19 patients [155] and also to date, there is no evidence for the potential adverse effect of these agents in COVID-19 patients. Furthermore, it is a tantalizing idea to explore whether administration of angiotensin-(Ang 1–7) can improve the clinical outcomes of COVID-19 patients. In principle, the use of Ang 1–7 and other strategies that exert endothelial protective effects against underlying endothelial dysfunction could alleviate COVID-19 symptoms and potentially reduce the severity of the disease. Finally, given that SARS-COV-2 induces endothelitis and aberrant angiogenesis [112], it is worth to investigate whether administration of anti-angiogenesis drugs in combination with other modalities could be considered in the treatment of COVID-19.

## 10. Conclusions

ARDS is the most critical clinical manifestation of COVID-19, which can be induced through immune response (i.e., cytokine storm) and vascular damage or combination of both pathways. Pre-existing endothelial dysfunction in patients with hypertension, diabetes, and obesity or aging combined with vascular damage-induced by SARS-CoV-2 could contribute to severe morbidly and mortality. Further investigation into the role of SARS-CoV-2 induced endothelial damage could shed new light into pathogenesis of COVID-19 and development of potential therapy.

## Figures and Tables

**Figure 1 cells-09-01652-f001:**
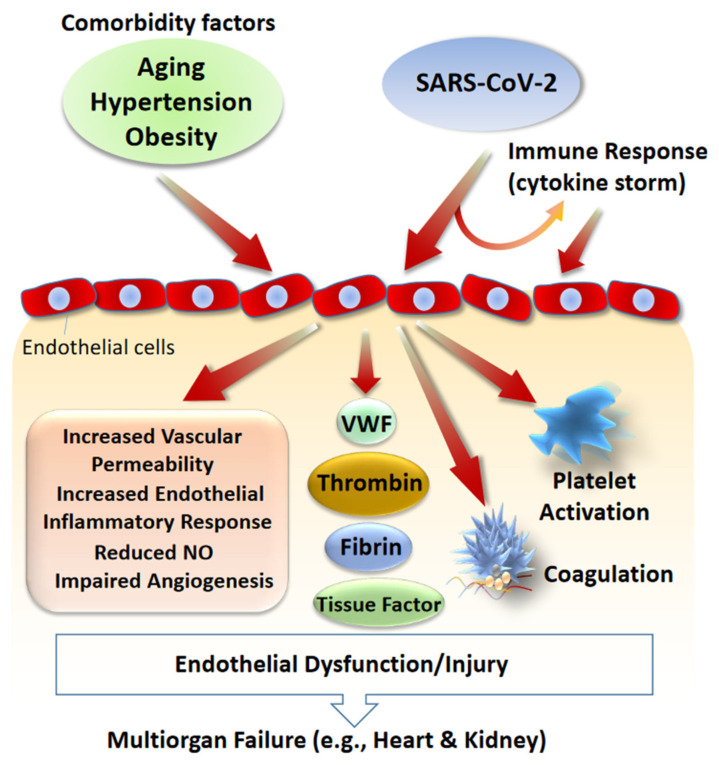
Role of comorbidity factors and SARS-CoV-2 in vascular dysfunction and vascular injury. Endothelial dysfunction is associated with aging and conditions such as hypertension and diabetes. SARS-CoV-2 can induce vascular damage directly or indirectly by stimulating immune response which results in excessive cytokine production (cytokine storm) which also can lead to vascular damage. SARS-CoV-2 induced vascular damage alone or in combination with pre-existing endothelial dysfunction can lead to multisystem organ failure and death. Key biochemical factors and cellular responses involved in the SARS-CoV-2 induced endothelial damage and endothelial dysfunction are shown.

**Figure 2 cells-09-01652-f002:**
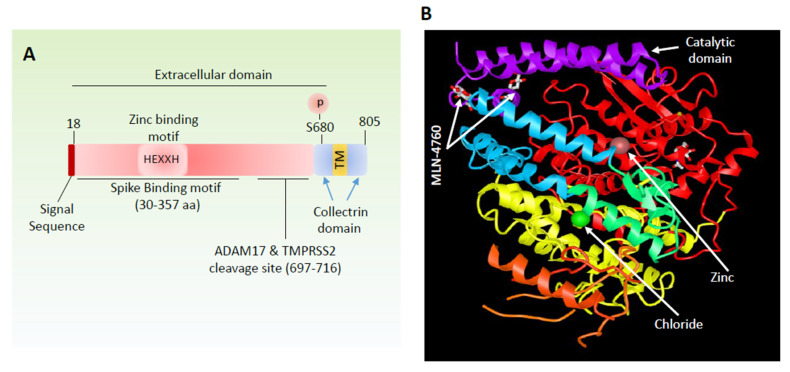
Schematic and domain structure of angiotensin-converting enzyme 2 (ACE2). (**A**) General domain information including, ion binding, proteolytic cleavage sites and S protein binding motif are shown. (**B**) Crystal structure of ACE2 and location of ion bindings and catalytic domain in complex with ACE2 inhibitor, MLN-476, is shown.

**Figure 3 cells-09-01652-f003:**
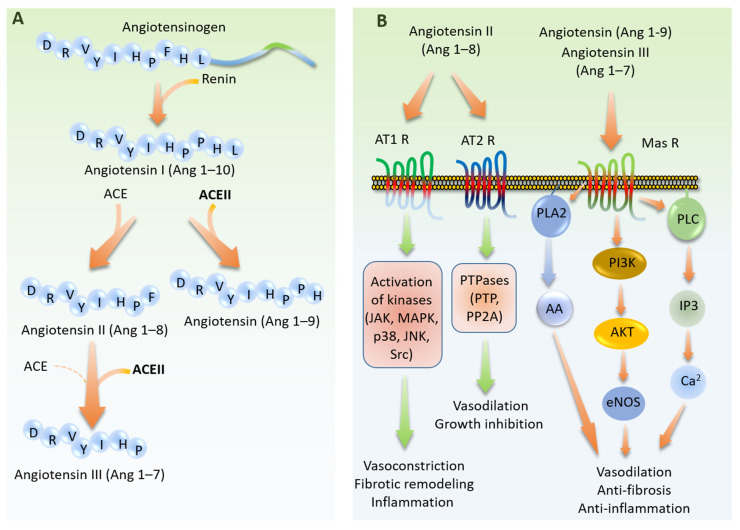
The physiological role angiotensin system in cardiovascular system. (**A**) Angiotensinogen is cleaved by renin and produces angiotensin I (Ang 1–10). Ang 1–10 is substrate for both ACE and ACE2. While ACE generates Ang 1–8, ACE2 cleaves Ang-10 and generates Ang 1–9. Ang 1–8 is a major substrate for ACE2 which produces Ang 1–7. (**B**) Ang (1–8) serves as a ligand for G-protein-coupled receptors (GPCRs), AT1R and AT2R. Activation of AT1R promotes vasoconstriction, fibrotic remodeling, and inflammation. Stimulation of AT2R leads to vasodilation and growth inhibition. On the other hand, peptides produced by ACE2 (Ang 1–9 and Ang 1–7) bind to GPCR, Mas receptor leading to its activation and promotes vasodilation, anti-fibrosis, and anti-inflammation effects. AA, Arachidonic acid.

**Figure 4 cells-09-01652-f004:**
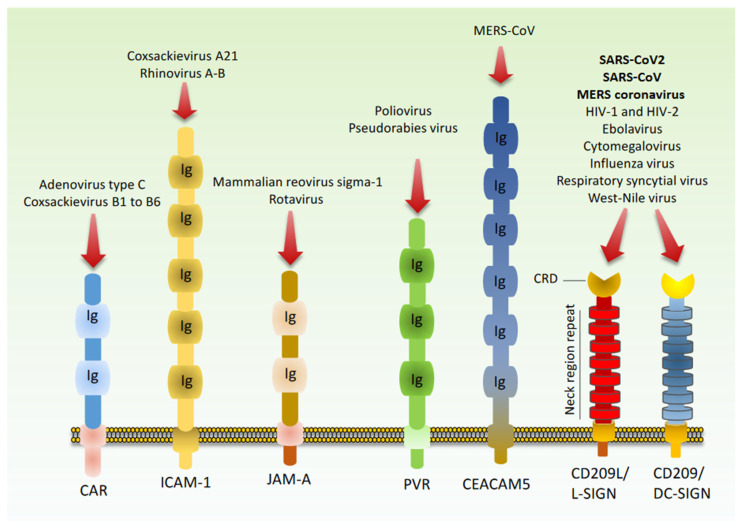
Cell adhesion of molecules are common receptors for viruses. Schematic of the most common cell adhesion molecules involved in virus recognition and viral cell entry. Coxsackievirus-adenovirus receptor (CAR) [127], intercellular adhesion molecule 1 (ICAM 1/CD54) [128], junctional adhesion molecule A (JAM-A) [129,130], and poliovirus receptor (PVR/CD155) [131]. MERS-CoV also employs carcinoembryonic antigen-related cell adhesion molecule 5 (CEACAM5) [132]. CLEC4M, (also known as L-SIGN and CD209L) and CAM, dendritic cell-specific ICAM-3 grabbing non-integrin (DC-SIGN, also called CD209) binds to Ebolavirus [133], Hepatitis C virus [134], human coronavirus 229E [135], Human cytomegalovirus/HHV-5 [136], Influenza virus [137], West-Nile virus [136], Japanese encephalitis virus [138], SARS-CoV [136,139,140], and SARS-CoV-2 [120].

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
