# Peer review of "COVID-19, Renin-Angiotensin System and Endothelial Dysfunction"

_cells, 2020, doi:10.3390/cells9071652_

Round 1

Reviewer 1 Report

The following review is timely considering the ongoing nature of the Covid-19 pandemic.  The authors nicely highlight the symptomology of Covid-19 as it relates to the cardiovascular system. They also review the role of ACE2 as the receptor for Covid-19 as well as the complex nature of ACE2 in disease progression as well as in the cardiovascular co-morbidities of obesity, hypertension, diabetes, and aging.

Specific Comments:

1.  While timely and well written, this review represents the third review this reviewer has reviewed albeit from different authors for different Journals, the theme appears to be exactly the same as a large number of reviews on the same topic. Thus, one issue common to the emerging reviews in publication is a large number of vagaries. It is clear that there is a lot we do not yet know, the present manuscript could be further refined. Examples include expanding on the precedent for other enzymes that “moonlight” as functional receptors and greater detail regarding the specific vascular and structural damage caused by Covid-19.

2.  As a vascular biologist, a lot of the vascular descriptions appear to be written by a non-vascular biologist.  Section 6 is vague in its descriptions, for example, “Endothelial dysfunction is defined by alterations in endothelium-regulating functions…” and exactly what endothelium functions are implied should be defined. “Vasoconstrictors and vasodilators” are also undefined, please provide examples. Also, in this section the authors state “some viruses…infect endothelial cells leading to vascular permeability.” – however, the vasculature is inherently permeable to some extent, the question really is how do viruses alter permeability.

3.  Please update Covid-19 infections and deaths worldwide since submission (introduction).

4.  It appears that Figure 1 figures have been placed with Figure 2 figure legend and vice versa. Please correct.

5.  Page 6, change “vast majority” to simply “majority.”

6.  Page 7, change “-COV” to “-CoV” where appropriate.

7.  Page 7, ICAM-1 previously defined, change intracellular adhesion molecule-1 to ICAM-1 here.

8.  Page 8, last sentence: Redundant as point previously discussed.

9.  Section 9, last two full paragraphs: Both could be improved as it is not clear how RAS drugs can be explored or exploited for COVID-19 patients. Also, discussion on nitric oxide appears to be an after thought (should have been discussed in part in Section 6) and “to produce NO synthesis” is not normal nomenclature in the field of vascular biology.

10.  Figure 1 legend: "role angiotensin system" should be "role of the renin-angiotensin system."

Author Response

Reviewer #1

  1.  While timely and well written…. It is clear that there is a lot we do not yet know, the present manuscript could be further refined. Examples include expanding on the precedent for other enzymes that “moonlight” as functional receptors and greater detail regarding the specific vascular and structural damage caused by Covid-19.

Thank you for the suggestion. The current revised manuscript expanded the moonlighting function of ACE2 and other molecules including cell adhesion molecules. 

  1.  Section 6 is vague in its descriptions, for example, “Endothelial dysfunction is defined by alterations in endothelium-regulating functions…” and exactly what endothelium functions are implied should be defined. “Vasoconstrictors and vasodilators” are also undefined, please provide examples. Also, in this section the authors state “some viruses…infect endothelial cells leading to vascular permeability.” – however, the vasculature is inherently permeable to some extent, the question really is how do viruses alter permeability.

In this revised manuscript we have fully define the endothelial dysfunction and provided adequate examples where endothelial dysfunction is linked to a particular human diseases. 

  1. 3.  Please update Covid-19 infections and deaths worldwide since submission (introduction).

We have updated the COVID-19 cases and deaths.

  1. 4.  It appears that Figure 1 figures have been placed with Figure 2 figure legend and vice versa. Please correct.

We apologize for the confusion. The figure legends now correctly correspond to appreciate figures.   

  1. 5.  Page 6, change “vast majority” to simply “majority.”

We have replaced “vast majority” to “majority”.

  1. 6.  Page 7, change “-COV” to “-CoV” where appropriate.

We apologize for the confusion. We have checked the entire manuscript for typos and others.  

  1. 7.  Page 7, ICAM-1 previously defined, change intracellular adhesion molecule-1 to ICAM-1 here.

We have revised the manuscript and manuscript now reads intracellular adhesion molecule-1 to ICAM-1.

  1. 8.  Page 8, last sentence: Redundant as point previously discussed.

We have deleted the sentence.

  1. 9.  Section 9, last two full paragraphs: Both could be improved as it is not clear how RAS drugs can be explored or exploited for COVID-19 patients. Also, discussion on nitric oxide appears to be an after thought (should have been discussed in part in Section 6) and “to produce NO synthesis” is not normal nomenclature in the field of vascular biology.

Thank you for the suggestion. We have revised our manuscript to reflect these considerations. 

  1.  Figure 1 legend: "role angiotensin system" should be "role of the renin-angiotensin system."

Thank you. Now, the figure legend reads “the role of the renin-angiotensin system”.

ave deleted the sentence.

  1. 9.  Section 9, last two full paragraphs: Both could be improved as it is not clear how RAS drugs can be explored or exploited for COVID-19 patients. Also, discussion on nitric oxide appears to be an after thought (should have been discussed in part in Section 6) and “to produce NO synthesis” is not normal nomenclature in the field of vascular biology.

Thank you for the suggestion. We have revised our manuscript to reflect these considerations. 

  1.  Figure 1 legend: "role angiotensin system" should be "role of the renin-angiotensin system."

Thank you. Now, the figure legend reads “the role of the renin-angiotensin system”.

Reviewer 2 Report

Amraie R and Rahimi N seek to review the molecular basis of SARS-CoV-2 infection, the roles of ACE2 and RAS signaling, and a possible link between the preexisting endothelial dysfunction and SARS-CoV-2 induced endothelial injury in COVID-19 associated mortality. This is an interesting review on a topical subject.

The authors have accurately interpreted and presented the relevant results. The manuscript's figures are easy to understand, however more figures should be included specially to explain the complex pathways and the proposed entry of the SARS-CoV-2 using ACE2 (e.g. on Angiotensin-converting enzyme 2 is a novel viral recognition receptor and Viral-host interaction activates spike protein through proteolytic activation). These would complement the message of the review and facilitate understanding of the pathways.

As mentioned above, the authors have accurately presented the relevant results, however the first half of the review lacks synthesis, analysis, and critique of the information. In the second half of the review, we do get to see more of what the authors think and what they propose, which makes it more interesting and useful to the readers.

Perhaps in the introduction, the authors could talk briefly about the symptoms associated with SARS-CoV-2.

Perhaps the authors could discuss papers like [Chen L, Li X, Chen M, Feng Y, Xiong C. The ACE2 expression in human heart indicates new potential mechanism of heart injury among patients infected with SARS-CoV-2. Cardiovasc Res. 2020;116(6):1097‐1100. doi:10.1093/cvr/cvaa078], which suggest that ACE2 was highly expressed in pericytes of adult human hearts, indicating an intrinsic susceptibility of heart to SARS-CoV-2 infection. Please discuss whether the protective effects of ACE2-local RAAS is applicable in this case

Minor

Consistency with abbreviations e.g. SARS-COV vs SARS-CoV, choose and stick to one.

Line 45, is it ‘three major genera,’ or should it be four major genera

There are some spelling mistakes

The same reference is repeated twice, 46 and 48

Author Response

Reviewer # 2

The authors have accurately interpreted and presented the relevant results. The manuscript's figures are easy to understand, however more figures should be included specially to explain the complex pathways and the proposed entry of the SARS-CoV-2 using ACE2 (e.g. on Angiotensin-converting enzyme 2 is a novel viral recognition receptor and Viral-host interaction activates spike protein through proteolytic activation). These would complement the message of the review and facilitate understanding of the pathways. However the first half of the review lacks synthesis, analysis, and critique of the information. In the second half of the review, we do get to see more of what the authors think and what they propose, which makes it more interesting and useful to the readers. Perhaps in the introduction, the authors could talk briefly about the symptoms associated with SARS-CoV-2.

Thank you for this insightful suggestion. We have expanded the introduction of the manuscript to include symptoms associated with COVID19 and also supplemented with an additional figure (Figure 1 in the revised manuscript) to further highlight the role of endothelial dysfunction in the pathogenesis of COVID-19.  

Perhaps the authors could discuss papers like [Chen L, Li X, Chen M, Feng Y, Xiong C. The ACE2 expression in human heart indicates new potential mechanism of heart injury among patients infected with SARS-CoV-2. Cardiovasc Res. 2020;116(6):1097‐1100. doi:10.1093/cvr/cvaa078], which suggest that ACE2 was highly expressed in pericytes of adult human hearts, indicating an intrinsic susceptibility of heart to SARS-CoV-2 infection. Please discuss whether the protective effects of ACE2-local RAAS is applicable in this case.

Thank you for the suggestion.  We have incorporated this recommendation in our revised manuscript (line 322) and reference # 132.  

Minor:

Consistency with abbreviations e.g. SARS-COV vs SARS-CoV, choose and stick to one.

We apologize for the confusion. We have corrected the entire manuscript for consistency.  

Line 45, is it ‘three major genera,’ or should it be four major genera.

Thank you for noticing this. It should be four major genera. We have corrected this typo.  

There are some spelling mistakes. The same reference is repeated twice, 46 and 48.

We have carefully edited the manuscript for the typos and corrected the references.

Reviewer 3 Report

The manuscript "COVID 19, angiotensin system and endothelial dysfunction" presented by Rahimi and Razie is a novel and interesting manuscript (revision). The manuscript is well written and has an adequate structure. The authors addressed the main molecular aspects involved in the angiotensin system and endothelial dysfunction. Considering the complex health crisis caused by covid 19, this manuscript will be of interest to researchers. However, I have the following comments.

I. Minor Comments:

1. In endothelial dysfunction, oxidative stress is a relevant aspect to initiate the inflammatory response. I suggest writing a short paragraph about it.
Suggested reference:

Impact of the Co-Administration of N-3 Fatty Acids and Olive Oil Components in Preclinical Nonalcoholic Fatty Liver Disease Models: A Mechanistic View. Nutrients. 2020; 12: 499.

2. Considering the antecedents presented by the authors (section 9), I suggest including a brief paragraph regarding the potential interventions parmacology in patients with covid-19 (active patients and recovered patients).

3. Improve the writing of the manuscript objective.

Author Response

Reviewer #3

  1. Minor Comments:
  2. 1. In endothelial dysfunction, oxidative stress is a relevant aspect to initiate the inflammatory response. I suggest writing a short paragraph about it. Suggested reference: Impact of the Co-Administration of N-3 Fatty Acids and Olive Oil Components in Preclinical Nonalcoholic Fatty Liver Disease Models: A Mechanistic View. Nutrients. 2020; 12: 499.

Thank you for suggestion. We have included a short paragraph (line 252-255) and also cited the suggested reference (reference #89).

2. Considering the antecedents presented by the authors (section 9), I suggest including a brief paragraph regarding the potential interventions parmacology in patients with covid-19 (active patients and recovered patients).

Although, this is out of the focus of this review, we briefly discussed this issue in the introduction of the manuscript.

  1. 3. Improve the writing of the manuscript objective.

Thank you for the suggestion. The revised manuscript now has more defined objectives as illustrated in the new figure (Figure 1) and inclusion of additional information in the introduction section of the manuscript.